# Chest pain and shortness of breath in cardiovascular disease: a prospective cohort study in UK primary care

Lauren A Barnett,[1] James A Prior,[1] Umesh T Kadam,[2] Kelvin P Jordan[1]

► Prepublication history and additional material are available. To view these files please visit the journal online (http://dx.doi.org/10.1136/bmjopen-2017-015857).

## ABSTRACT

**Objective**  To determine characteristics associated with monthly chest pain and shortness of breath (SoB) during activity in cardiovascular disease (CVD) and trajectories of these symptoms over 10 months.

**Study design and setting**  Baseline questionnaire was sent to patients aged ≥40 years from 10 UK general practices. Responders were sent monthly questionnaires for 10 months. For patients with CVD (ischaemic heart disease and heart failure), the association of sociodemographic characteristics, pain elsewhere and anxiety and depression with monthly reports of chest pain and SoB during activity were determined using multilevel, multinomial logistic regression. Common symptom trajectories were determined using dual trajectory latent class growth analysis.

**Results**  661 patients with CVD completed at least 5 monthly questionnaires. Multiple other pain sites (relative risk ratio: 4.03; 95% CI 1.64 to 9.91) and anxiety or depression (relative risk ratio: 3.31; 95% CI 1.89 to 5.79) were associated with reporting weekly chest pain. Anxiety or depression (relative risk ratio: 4.10; 95% CI 2.72 to 6.17), obesity (relative risk ratio: 2.53; 95% CI 1.49 to 4.30), older age (80+: relative risk ratio: 2.51; 95% CI 1.19 to 5.26), increasing number of pain sites (4+: relative risk ratio: 4.64; 95% CI 2.35 to 9.18) and female gender (relative risk ratio: 1.81; 95% CI 1.20 to 2.75) were associated with reporting weekly SoB. Eight symptom trajectories were identified, with SoB symptoms more common than chest pain.

**Conclusions**  Potentially modifiable characteristics are associated with the experience of chest pain and SoB. Identified symptom trajectories may facilitate tailored care to improve outcomes in patients with CVD.

## Strengths and limitations of this study

► This was a large study of primary care patients with diagnosed cardiovascular (CVD) disease.
► It is the first study to examine the course of CVD symptoms based on monthly self-reported data.
► There was attrition, with many patients not completing sufficient monthly questionnaires to be included in the analysis.
► Due to the nature of the questionnaire data, we were unable to identify the number or the duration of the monthly symptom episodes.

Though directly associated with CVD, chest pain and SoB may also be experienced due to non-cardiac reasons. In a US sample of 10 881 participants with no prevalent CVD or pulmonary conditions, 22% reported SoB,[8] and SoB and chest pain have been shown to be associated with other patient characteristics such as age, gender and depression.[9] Understanding potentially modifiable patient characteristics associated with the symptoms of chest pain and SoB (whether related to CVD pathology) in patients diagnosed with CVD may present additional opportunities to direct management to reduce the risk of disease progression and prolonged symptom experience and to improve outcomes in patients with CVD.

Symptoms of chest pain and SoB are not typically isolated events. Assessing how patients' symptoms co-occur and change over time may help identify common patterns (trajectories) of these symptoms that may be related to long-term outcomes. However, how the experience of chest pain and SoB in patients with CVD varies over time remains unclear. Other trajectories of symptoms have previously been determined to aid understanding of the course of morbidity, for example, in back pain.[10] Understanding the different trajectories of patient symptoms can be beneficial when trying to optimise healthcare resources, for example, through stratified care.[11] Such methods may allow

## INTRODUCTION

Chest pain and shortness of breath (SoB) are common symptoms of cardiovascular disease (CVD) seen in UK primary care. The incidence of chest pain ranges between 13 and 20 per 1000 person-years,[1–3] with chest pain consultation being a risk factor for future cardiovascular morbidity and mortality.[2 4] SoB is a frequent symptom of heart failure (HF) and angina.[5 6] Patients with CVD may experience and consult for chest pain and SoB separately, while many patients will experience both together.[7]

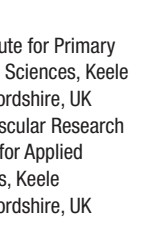

[1]Research Institute for Primary Care and Health Sciences, Keele University, Staffordshire, UK
[2]Keele Cardiovascular Research Group, Institute for Applied Clinical Sciences, Keele University, Staffordshire, UK

**Correspondence to**
Dr. James A Prior;
j.a.prior@keele.ac.uk

better targeting of interventions, potentially earlier or to those with greater need.

The objectives of our study, in a primary care population with CVD, were; (1) to determine the association of patient characteristics with the monthly experience of chest pain or SoB during activity and (2) to determine common trajectories of chest pain and SoB symptoms over time.

## METHODS

### Study design

This analysis uses data from the Comorbidity Cohort (2C) study. A prospective cohort study, this was designed to investigate the interaction between CVD and osteoarthritis (OA) comorbidity over 12 months. Full details of the 2C study are reported elsewhere.[12-14] In brief, the 2C study sampled patients with recorded cardiovascular morbidity and/or OA, aged 40 years or older, plus a reference group without CVD or OA, from the electronic health records of 10 UK general practices. All patients fulfilling the study inclusion criteria were mailed a baseline questionnaire, which included measures of general health as well as CVD and OA specific measures. Those who completed the baseline questionnaire and consented to further follow-up were mailed shorter monthly health questionnaires for the following 10 months and a larger questionnaire at 12 months.

### Study population

This analysis focuses only on those patients with recorded CVD (with or without comorbid OA) of which there were 2858 sent a baseline questionnaire. These patients were identified through a primary care record of ischaemic heart disease (IHD) or HF using Read codes (commonly used to record morbidity in UK primary care) within a 3-year period prior to the baseline questionnaire, starting from November 2006. IHD was identified by Read codes starting G3 ('Ischaemic heart disease'), and HF was identified by Read codes starting G58 ('Heart failure') and codes related to the New York Heart Association classification (codes 662f-i).

### Outcome measures

The primary outcomes were monthly self-reported measures of two CVD-related symptoms: (1) chest pain during activity and (2) SoB during activity. A single-item question was used for each symptom, both based on existing, valid outcome measures (the Seattle Angina Questionnaire[15] and the Kansas City Cardiomyopathy Questionnaire)[16]. Patients were asked 'How often (if at all) have you had chest pain during activity over the past 4 weeks?'. The same question structure was used for SoB. Response options for these questions were 'Not at all', 'For 1 week or less', 'For 2 weeks', 'For 3 weeks' and 'For 4 weeks. For analysis, these were categorised into: 'not at all', 'for 1–3 weeks' and 'for 4 weeks', representing no, episodic and persistent monthly symptoms, respectively.

### Covariates

Covariates measured in the baseline questionnaire were age, gender, body mass index (BMI; based on self-reported height and weight), general physical health, pain elsewhere and anxiety and depression. BMI was categorised as: normal weight (<25), overweight (25–30) or obese (30+). General physical health was measured using the 12-item Short-Form Health Survey.[17] From this, the Physical Component Summary (PCS) score was derived; this is normalised to the US general population with scores below 50 reflecting worse physical health than for the general population. Anxiety and depression were measured using the Hospital Anxiety and Depression Scale with the possible score ranging from 0 (best) to 21 (worst) for anxiety and for depression.[18] Given the strong association between anxiety and depression, we combined the two scales and defined those with borderline or probable anxiety or depression (based on standard cut-offs) as anxious or depressed. We determined the extent of pain elsewhere at baseline as a count of 10 self-reported areas of pain (neck, shoulder, elbow, hand, back, hip, knee, foot, abdominal and headache). This was categorised into none, 1–3, and 4 or more sites. Lacey et al[19] suggested the number of self-reported pain sites is more important than the pain location.[19]

Baseline chest pain and SoB were measured using yes/no responses to items enquiring about any chest pain or tightness in the past 4 weeks, and any SoB in the past 4 weeks.

### Statistical analyses

The statistical analysis was performed in two phases. The first phase examined associations between baseline sample characteristics and monthly chest pain and SoB experienced during activity. The second phase clustered patients into groups based on the most common patterns (trajectories) of chest pain and SoB experienced over the 10-month follow-up time period. Responders who had completed chest pain or SoB items on at least 5 of the 10 monthly questionnaires were included in the analyses.

#### Phase one: associations between baseline characteristics, chest pain and SoB

A multilevel, multinomial logistic regression model (monthly measurements nested within respondents) was used to determine associations between the baseline sample characteristics (age, gender, BMI, general physical health, pain elsewhere, anxiety and depression) and monthly chest pain during activity (1–3 weeks vs none and 4+ weeks vs none). Associations were initially unadjusted, followed by adjustment for covariates. PCS score was not included in the multivariable model due to strong association with age and pain elsewhere. Interactions between time and covariates that were found to be statistically significant were incorporated into the model to assess whether their effects changed over the time period. The same analysis was then repeated for SoB.

**Table 1**  Sample characteristics, n (%)

| | Invited at baseline | Baseline responders | At least 1 monthly response | At least 2 monthly responses* | At least 5 monthly responses† |
|---|---|---|---|---|---|
| n | 2858 | 1696 | 871 | 777 | 661 |
| Condition group | | | | | |
| IHD | 2526 (88) | 1501 (89) | 783 (90) | 701 (90) | 603 (91) |
| HF | 332 (12) | 195 (11) | 88 (10) | 76 (10) | 58 (9) |
| Age | | | | | |
| <60 | 401 (14) | 172 (10) | 103 (12) | 89 (11) | 75 (11) |
| 60–69 | 782 (27) | 463 (27) | 265 (30) | 241 (31) | 201 (30) |
| 70–79 | 987 (35) | 652 (38) | 339 (39) | 304 (39) | 274 (41) |
| 80+ | 688 (24) | 409 (24) | 164 (19) | 143 (18) | 111 (17) |
| Gender | | | | | |
| Male | 1712 (60) | 1059 (62) | 574 (66) | 514 (66) | 447 (68) |
| Female | 1146 (40) | 637 (38) | 297 (34) | 263 (34) | 214 (32) |
| Baseline chest pain | N/A | 689 (44) | 359 (44) | 318 (43) | 266 (42) |
| Baseline SoB | N/A | 942 (58) | 495 (58) | 430 (57) | 354 (54) |

*Population for sensitivity analysis.
†Population for main analysis.
HF, heart failure; IHD, ischaemic heart disease; SoB, shortness of breath.

Sensitivity analyses were performed using respondents who completed the chest pain and SoB items on at least 2 monthly questionnaires and by using multiple imputation (with 50 imputations) to impute the missing monthly questionnaire answers and any missing covariate data.

### Phase two: chest pain and SoB cluster trajectories

Dual trajectory latent class growth analysis (LCGA) was carried out to group (cluster) respondents into the most common trajectories of self-reported chest pain and SoB over the 10 months using the repeated monthly measures of the two symptoms.[20] Each cluster, therefore, represents a common trajectory of both chest pain and SoB over the 10 months. A fundamental assumption behind dual trajectory LCGA is that the two symptoms, in this case, chest pain and SoB, are correlated. The time order of the measurements were taken into account when deriving the trajectories within a cluster by using quadratic growth curves.

There is no definitive method of deciding on the most appropriate number of clusters using statistical goodness of fit measures. We initially used Akaike's information criterion, Bayesian information criterion (BIC) and the adjusted BIC, and for each of these, the model with the lowest goodness of fit value indicates the optimal number of clusters. The final decision on the optimal number of clusters was determined by a combination of statistical information, the size and distinctiveness of the clusters and how well the profile (monthly responses of chest pain and SoB) of respondents matched the cluster . Respondents were allocated to the cluster for which they had the highest posterior probability of belonging. The mean of these posterior probabilities for respondents assigned to each cluster (ie, cluster-specific average posterior probabilities (AvePP)) was determined. An AvePP greater than 0.7 suggests clear classification of participants into that cluster.[21] Cluster-specific probabilities of reporting of chest pain and SoB for each month allow profiles of the trajectory of CVD symptoms to be developed for each cluster. The clusters were then descriptively compared on the baseline covariates described earlier.

As a sensitivity analysis, LCGA was repeated using participants completing the chest pain and SoB items on at least 2 monthly questionnaires. A further sensitivity analysis used multiple imputation to impute clusters for all baseline CVD respondents, with 50 imputations, and using baseline age, gender, BMI, anxiety, depression, PCS score, number of pain sites elsewhere and baseline chest pain, SoB and HF diagnosis in the imputation. All analysis was done using Stata V.14, Mplus V.7.3 and MLwiN V.2.35.

### RESULTS

#### Patient demographics

Two thousand and eight hundred and fiftyeight patients with CVD were mailed the baseline questionnaire, of which 1696 (59%) responded. Six hundred and sixtyone (39%) baseline responders answered the chest pain and SoB items on between 5 and 10 monthly questionnaires and were included in the main analyses reported here. Compared with all other baseline responders, those in the analysis were younger (mean difference 2.3 years), had a higher proportion of males (68% vs 62%) and had a slightly lower prevalence of chest pain (42% vs 44%) and SoB (54% vs 58%) at baseline (table 1).

**Table 2** Associations between baseline characteristics and monthly chest pain (n=661)

| Baseline characteristics | Monthly chest pain | | | |
| --- | --- | --- | --- | --- |
| | 1–3 weeks versus none | | 4 weeks versus none | |
| | Unadjusted RRR (95% CI) | Adjusted* RRR (95% CI) | Unadjusted RRR (95% CI) | Adjusted* RRR (95% CI) |
| Male | 1 | 1 | 1 | 1 |
| Female | **1.50 (1.08 to 2.08)** | 1.20 (0.77 to 1.54) | 1.48 (0.90 to 2.43) | 1.29 (0.73 to 2.28) |
| Age | | | | |
| <60 | 1 | 1 | 1 | 1 |
| 60–69 | 1.35 (0.81 to 2.23) | 1.23 (0.74 to 2.06) | 0.64 (0.31 to 1.33) | 0.65 (0.28 to 1.49) |
| 70–79 | 1.39 (0.85 to 2.27) | 1.41 (0.86 to 2.34) | 0.49 (0.24 to 0.99) | 0.49 (0.21 to 1.11) |
| 80+ | 1.25 (0.68 to 2.29) | 1.18 (0.64 to 2.16) | 0.42 (0.17 to 1.03) | 0.47 (0.17 to 1.28) |
| BMI | | | | |
| Normal | 1 | 1 | 1 | 1 |
| Overweight | 0.88 (0.60 to 1.28) | 0.79 (0.54 to 1.15) | 0.93 (0.53 to 1.65) | 0.80 (0.42 to 1.51) |
| Obese | 0.95 (0.63 to 1.45) | 0.67 (0.43 to 1.03) | 1.77 (0.94 to 3.34) | 0.74 (0.36 to 1.54) |
| PCS score (per unit) | **0.95 (0.93 to 0.96)** | – | **0.89 (0.87 to 0.91)** | – |
| Not anxious or depressed | 1 | 1 | 1 | 1 |
| Anxious or depressed | **2.55 (1.85 to 3.51)** | **2.32 (1.66 to 3.24)** | **5.42 (3.19 to 9.19)** | **3.31 (1.89 to 5.79)** |
| Pain sites | | | | |
| None | 1 | 1 | 1 | 1 |
| 1–3 | **3.38 (1.97 to 5.81)** | **3.01 (1.77 to 5.13)** | 1.19 (0.50 to 2.81) | 1.24 (0.51 to 3.01) |
| 4+ | **5.66 (3.32 to 9.66)** | **3.89 (2.25 to 6.74)** | **6.16 (2.69 to 14.14)** | **4.03 (1.64 to 9.91)** |
| Between-person variance | – | 2.39 | – | 7.07 |

Bold values mean statistically significant
*Adjusted for all presented variables.
BMI, body mass index; PCS, Physical Component Summary; RRR, relative risk ratio.

During the 10-month follow-up period, prevalence of chest pain during activity in all 4 weeks ranged from 11% to 13% by month and that of chest pain for 1–3 weeks from 23% to 28%. Monthly prevalence of SoB during activity in all 4 weeks ranged from 21% to 26% and SoB for 1–3 weeks from 34% to 37%.

Phase one: associations between baseline characteristics and chest pain

Unadjusted analysis showed increasing number of pain sites elsewhere, being anxious or depressed and having a worse PCS score were associated with an increase in risk of reporting chest pain in all 4 weeks in a month.

After adjustment for all covariates, an increasing number of pain sites (relative risk ratio (RRR) for 4+ vs none: 4.03, 95% CI 1.64 to 9.91) and being anxious or depressed (RRR: 3.31, 95% CI 1.89 to 5.79) were associated with reporting chest pain in all 4 weeks in the previous month (table 2). The interactions of anxiety or depression and number of pain sites at baseline with time were not statistically significant.

Phase one: associations between baseline characteristics and SoB

Unadjusted analysis showed more pain sites elsewhere, being anxious or depressed, obesity, female gender and having a worse PCS score were associated with an increase in risk of reporting SoB during activity in all 4 weeks in the last month.

After adjustment for all covariates, increasing number of pain sites (RRR for 4+ vs none: 4.64, 95% CI 2.35 to 9.18), being anxious or depressed (RRR: 4.10, 95% CI 2.72 to 6.17) and obesity (RRR: 2.53, 95% CI 1.49 to 4.30) were associated with increased risk of reporting SoB in each of the previous 4 weeks, as were female gender and being over 80 years of age. There were no statistically significant interactions with time (table 3).

Phase one: -sensitivity analyses

Sensitivity analyses including those responding on at least 2 monthly questionnaires (n=777) and using multiple imputation did not change the findings (see online supplementary tables 1–4).

Phase two:- trajectories of chest pain and SoB

The goodness of fit measures from the LCGA did not clearly suggest an optimal number of clusters of trajectories of chest pain and SoB, although their values levelled off from the model with eight clusters (table 4). The six-cluster model had a lowest observed cluster size of 7% (n=47) of participants, compared with 4% (24

**Table 3** Associations between baseline characteristics and monthly shortness of breath (n=661)

| Baseline characteristics | Monthly shortness of breath | | | |
| --- | --- | --- | --- | --- |
| | 1–3 weeks versus none | | 4 weeks versus none | |
| | Unadjusted RRR (95% CI) | Adjusted* RRR (95% CI) | Unadjusted RRR (95% CI) | Adjusted* RRR (95% CI) |
| Male | 1 | 1 | 1 | 1 |
| Female | **1.84 (1.36 to 2.49)** | **1.46 (1.07 to 2.01)** | **2.58 (1.78 to 3.75)** | **1.81 (1.20 to 2.75)** |
| Age | | | | |
| <60 | 1 | 1 | 1 | 1 |
| 60–69 | 1.30 (0.81 to 2.07) | 1.18 (0.74 to 1.90) | 1.10 (0.62 to 1.95) | 0.92 (0.49 to 1.71) |
| 70–79 | **1.80 (1.14 to 2.83)** | **1.83 (1.15 to 2.90)** | 1.40 (0.80 to 2.44) | 1.72 (0.94 to 3.17) |
| 80+ | **1.91 (1.10 to 3.31)** | **2.47 (1.41 to 4.34)** | 1.53 (0.77 to 3.02) | **2.51 (1.19 to 5.26)** |
| BMI | | | | |
| Normal | 1 | 1 | 1 | 1 |
| Overweight | 1.25 (0.88 to 1.76) | 1.28 (0.90 to 1.81) | 1.02 (0.66 to 1.58) | 1.03 (0.64 to 1.65) |
| Obese | **1.77 (1.20 to 2.62)** | **1.67 (1.12 to 2.51)** | **2.93 (1.80 to 4.78)** | **2.53 (1.49 to 4.30)** |
| PCS score (per unit) | **0.93 (0.92 to 0.94)** | – | **0.87 (0.86 to 0.89)** | – |
| Not anxious or depressed | 1 | 1 | 1 | 1 |
| Anxious or depressed | **3.04 (2.26 to 4.08)** | **2.33 (1.70 to 3.19)** | **5.36 (3.67 to 7.82)** | **4.10 (2.72 to 6.17)** |
| Pain sites | | | | |
| None | 1 | 1 | 1 | 1 |
| 1–3 | **2.23 (1.41 to 3.54)** | **1.98 (1.23 to 3.17)** | **4.06 (2.20 to 7.50)** | **3.50 (1.80 to 6.80)** |
| 4+ | **4.31 (2.74 to 6.80)** | **3.05 (1.88 to 4.97)** | **8.90 (4.87 to 16.29)** | **4.64 (2.35 to 9.18)** |
| Between-person variance | – | 2.09 | – | 3.72 |

Bold values mean statistically significant.
*Adjusted for all presented variables.
BMI, body mass index; PCS, Physical Component Summary; RRR, relative risk ratio.

participants) for the seven-cluster model, and 3% (20 participants) for the model with eight clusters. Based on the goodness of fit measures, the size of clusters and their interpretation, the eight-cluster model was preferred (table 5).

Patients allocated to the largest cluster (n=166, 25%) generally reported no chest pain or SoB symptoms (with monthly probability 0.95 and above) in all the monthly questionnaires. Across clusters, in those reporting symptoms, SoB was more common than chest pain. One cluster

**Table 4** Goodness of fit of the dual trajectory models with differing numbers of clusters

| Clusters | AIC | BIC | Adjusted BIC | Smallest class (overall) (%) | AvePP* (range) |
| --- | --- | --- | --- | --- | --- |
| 1 | 23 241.43 | 23 277.38 | 23 251.98 | | |
| 2 | 17 999.11 | 18 066.52 | 18 018.89 | | |
| 3 | 16 185.44 | 16 284.30 | 16 214.45 | 16 | 0.98–0.98 |
| 4 | 15 088.50 | 15 218.82 | 15 126.74 | 15 | 0.95–0.99 |
| 5 | 14 310.02 | 14 471.80 | 14 357.50 | 9 | 0.94–0.99 |
| 6 | 14 018.94 | 14 212.17 | 14 075.64 | 7 | 0.93–0.98 |
| 7 | 13 820.48 | 14 045.17 | 13 886.42 | 4 | 0.92–0.98 |
| 8 | 13 701.00 | 13 957.15 | 13 776.17 | 3 | 0.89–0.99 |
| 9 | 13 628.19 | 13 915.79 | 13 712.59 | 0.1 | 0.88–1.00 |

*AvePP for participants allocated to cluster.
AIC, Akaike information criterion; AvePP, average posterior probability; BIC, Bayesian information criterion.

**Table 5** Summary of the clusters (eight-cluster model)

| Description | Cluster 1 No symptoms | Cluster 2 Infrequent SoB | Cluster 3 Occasional chest pain | Cluster 4 Occasional SoB | Cluster 5 Occasional chest pain and SoB | Cluster 6 Frequent chest pain and SoB | Cluster 7 Persistent SoB | Cluster 8 Persistent chest pain and SoB |
|---|---|---|---|---|---|---|---|---|
| Number of patients in cluster (%) | 166 (25) | 93 (14) | 20 (3) | 83 (13) | 125 (19) | 75 (11) | 52 (8) | 47 (7) |
| Range of monthly probabilities of extent of chest pain/SoB | | | | | | | | |
| Chest pain | | | | | | | | |
| None | 0.95–1.00 | 0.86–0.91 | 0.13–0.25 | 0.85–0.91 | 0.22–0.27 | 0.06–0.10 | 0.85–0.92 | <0.01 |
| Up to 3 weeks | <0.01–0.05 | 0.09–0.14 | 0.66–0.68 | 0.09–0.15 | 0.64–0.67 | 0.57–0.66 | 0.08–0.15 | 0.04–0.10 |
| 4 weeks | <0.01 | <0.01 | 0.10–0.20 | <0.01 | 0.09–0.11 | 0.24–0.37 | <0.01 | 0.90–0.96 |
| SoB | | | | | | | | |
| None | 0.96–0.99 | 0.52–0.65 | 0.76–0.93 | 0.13–0.19 | 0.15–0.16 | 0.01–0.02 | <0.01 | <0.01 |
| Up to 3 weeks | 0.01–0.04 | 0.34–0.46 | 0.07–0.23 | 0.72–0.74 | 0.73–0.73 | 0.33–0.45 | 0.11–0.16 | 0.03–0.12 |
| 4 weeks | <0.01 | 0.01–0.02 | <0.01 | 0.09–0.13 | 0.11–0.12 | 0.53–0.66 | 0.84–0.88 | 0.87–0.97 |
| Median (IQR) % of months with chest pain/SoB | | | | | | | | |
| Chest pain | | | | | | | | |
| None | 100 (100,100) | 90 (80,100) | 21 (0,50) | 100 (80,100) | 20 (0,40) | 0 (0,14) | 100 (81,100) | 0 (0,0) |
| Up to 3 weeks | 0 (0,0) | 10 (0,20) | 60 (40,79) | 0 (0,20) | 70 (56,89) | 57 (40,70) | 0 (0,13) | 0 (0,13) |
| 4 weeks | 0 (0,0) | 0 (0,0) | 0 (0,16) | 0 (0,0) | 0 (0,11) | 33 (17,50) | 0 (0,0) | 100 (83,100) |
| SoB | | | | | | | | |
| None | 100 (100,100) | 63 (50,75) | 100 (79,100) | 11 (0,25) | 10 (0,20) | 0 (0,0) | 0 (0,0) | 0 (0,0) |
| Up to 3 weeks | 0 (0,0) | 30 (20,44) | 0 (0,17) | 80 (70,90) | 80 (63,100) | 33 (20,50) | 0 (0,20) | 0 (0,13) |
| 4 weeks | 0 (0,0) | 0 (0,0) | 0 (0,0) | 0 (0,20) | 0 (0,11) | 60 (50,78) | 90 (79,100) | 100 (86,100) |

SoB, shortness of breath.

of patients reported infrequent breathlessness symptoms (reporting SoB for up to 3 weeks on a median of 3 of the 10 monthly questionnaires: n=93, 14%), another cluster of patients reported occasional breathlessness symptoms (reporting SoB for up to 3 weeks in a median of 8 out of the 10 months: n=83, 13%) and a third cluster of patients reported persistent breathlessness symptoms (reporting SoB in all of the last 4 weeks in all monthly questionnaires: n=52, 8%), but people in these three clusters rarely reported chest pain. The smallest cluster reported occasional chest pain, but rarely reported SoB (n=20, 3%). Two clusters reported occasional (n=125, 19%) or frequent (no months without symptoms, n=75, 11%) pain alongside breathlessness. The most severe group of patients reported persistent SoB and pain over the 10 months, that is, reported SoB and pain for all 4 weeks in all monthly questionnaires (n=47, 7%).

Patients in the clusters with both pain and SoB that was frequent or persistent had poorer baseline comorbid symptoms (anxiety and depression, and pain elsewhere) and worse physical health in general (table 6). Levels of obesity were higher in those with persistent SoB (with or without pain). The most severe group were younger than the other symptomatic groups.

Sensitivity analysis, including respondents with at least 2 monthly responses, showed a similar distribution of participants across clusters (see online supplementary table 5). Multiple imputation of cluster membership for all baseline respondents reduced the percentage in the no symptoms cluster (25% to 22%) and increased the percentage in the frequent pain and SoB cluster (11% to 14%) (see online supplementary table 6).

## DISCUSSION

### Summary

Using a prospective cohort study of patients with CVD, we found that potentially modifiable characteristics (weight, psychological health and number of painful body sites) influence the experience of both occasional and persistent chest pain and SoB during activity, and that eight distinct symptom trajectories are apparent, with breathlessness symptoms more common than chest pain.

Three of our identified clusters consisted of patients with stable conditions. The most common cluster (containing a quarter of our sample) reported no symptoms every month. By contrast, one cluster reported persistent chest pain and SoB every month, and another reported persistent SoB without chest pain every month. Symptoms of patients in the remaining clusters fluctuated more. We did not observe a cluster who improved or worsened over the 10 months, which may reflect the prevalent rather than incident nature of the cohort.

Though chest pain and SoB may be considered a distinctive symptom of acute CVD conditions, such as myocardial infarction, a quarter of our prevalent CVD sample did not experience these symptoms at all, similar to a study of community recruited patients with HF from the UK.[22] Chest pain and SoB commonly co-occurred, but SoB was more common and was reported without chest pain in three of our clusters (containing over a third of our cohort). We found a strong association between poor psychological health and both chest pain and SoB. Our findings are supported by previous cross-sectional studies that showed a correlation between SoB and depression both in US and Chinese secondary care HF samples.[23 24] Approximately two-thirds of patients with CVD in the most severe symptom clusters (frequent or persistent chest pain and SoB) had anxiety or depression. These results are supported by longitudinal research that reported poorer treatment outcomes in angina patients with chest pain and depression, compared with those without depression[25] and that in angina patients with persistent symptoms, long-term anxiety and depression were more likely.[26] Our research shows that the psychological health of patients with CVD, especially those with frequent chest pain and SoB, is poor. However, at present, our findings only indicate the association between chest pain, SoB and poor mental health. It remains unclear whether interventions aimed at anxiety or depression would improve these CVD symptoms, though this does present a potential avenue for future research.

Patients with CVD who were obese at baseline were more likely to experience SoB compared with those of normal weight. This in line with other research and also demonstrates potential to improve this common CVD symptom. Bernhardt and Babb[27] found that moderate weight loss in women who were obese at baseline was effective at reducing breathlessness during exertion.

We have shown that chest pain and SoB are associated with anxiety and depression and pain elsewhere, and addressing these characteristics may benefit both symptoms. Clark *et al* reported that improving (or diminishing) SoB after 2 years depended on whether pain resolved or diminished respectively.[7] As a third of our sample reported both chest pain and SoB, these present a considerable proportion of patients with CVD where future research may look to develop interventions towards a singular problem (eg, anxiety), which may also benefit the two symptoms.

Our clusters provide distinct patterns of symptom experience over time within a CVD population. Three-quarters of patients with CVD experienced either SoB or chest pain; however, approximately a third of our sample (35%) reported SoB only, either infrequently (14%), occasionally (13%) or persistently (8%). These may represent a cluster of patients at greater risk of poor outcomes. In a large sample of US participants with coronary artery disease at baseline, the mortality rates were twice as high in patients with SoB compared with those with other symptoms and four times greater than in those with no symptoms at presentation.[28] Symptoms such a chest pain and SoB may act as prognostic indicators for outcomes resulting in[8] or from CVD,[28] and therefore our findings present a range of likely trajectories to which future research may tailor interventions.

**Table 6** Comparison between clusters on baseline sociodemographic and general health measures, n (%) unless otherwise stated

| Description | Cluster 1 No symptoms | Cluster 2 Infrequent SoB | Cluster 3 Occasional chest pain | Cluster 4 Occasional SoB | Cluster 5 Occasional chest pain and SoB | Cluster 6 Frequent chest pain and SoB | Cluster 7 Persistent SoB | Cluster 8 Persistent chest pain and SoB |
|---|---|---|---|---|---|---|---|---|
| Number of patients in cluster | 166 (25) | 93 (14) | 20 (3) | 83 (13) | 125 (19) | 75 (11) | 52 (8) | 47 (7) |
| Female | 33 (20) | 31 (33) | 5 (25) | 29 (35) | 43 (34) | 31 (41) | 22 (42) | 20 (43) |
| Age | | | | | | | | |
| <60 | 26 (16) | 7 (8) | 4 (20) | 9 (11) | 11 (9) | 6 (8) | 2 (4) | 10 (21) |
| 60–69 | 59 (36) | 28 (30) | 5 (25) | 17 (20) | 42 (34) | 17 (23) | 17 (33) | 16 (34) |
| 70–79 | 59 (36) | 40 (43) | 9 (45) | 37 (45) | 53 (42) | 34 (45) | 24 (46) | 18 (38) |
| 80+ | 22 (13) | 18 (19) | 2 (10) | 20 (24) | 19 (15) | 18 (24) | 9 (17) | 3 (6) |
| BMI group | | | | | | | | |
| Normal weight | 50 (31) | 24 (28) | 10 (50) | 16 (20) | 30 (26) | 17 (25) | 10 (20) | 9 (20) |
| Overweight | 81 (50) | 47 (55) | 7 (35) | 39 (49) | 57 (49) | 29 (43) | 16 (31) | 14 (32) |
| Obese | 31 (19) | 15 (17) | 3 (15) | 25 (31) | 29 (25) | 22 (32) | 25 (49) | 21 (48) |
| Mean PCS (SD) | 47.1 (10.26) | 39.5 (10.89) | 39.8 (11.4) | 38.3 (10.43) | 36.93 (9.71) | 28.5 (8.91) | 31.0 (9.99) | 24.6 (7.05) |
| Anxious or depressed | 17 (11) | 20 (22) | 6 (30) | 32 (40) | 51 (42) | 48 (69) | 20 (38) | 29 (63) |
| Pain sites | | | | | | | | |
| None | 38 (23) | 15 (16) | 2 (10) | 5 (6) | 8 (6) | 3 (4) | 4 (8) | 1 (2) |
| 1–3 sites | 83 (50) | 39 (42) | 8 (40) | 30 (36) | 45 (36) | 19 (25) | 25 (48) | 10 (21) |
| 4+ sites | 45 (27) | 39 (42) | 10 (50) | 48 (58) | 72 (58) | 53 (71) | 23 (44) | 36 (77) |

BMI, body mas index; PCS, Physical Component Summ ary; SoB, shortness of breath.

## Strengths and limitations

The strength of our study lies in its prospective cohort design that has used a large CVD sample from primary care to gain an insight into the monthly experience of chest pain and SoB symptom in this sample. This study was conducted within a UK primary care setting using monthly surveys from patients with a diagnosis of CVD, identified through electronic health records from 10 general practices. Since these patients were identified from general practices across North Staffordshire, Stoke-on-Trent and Cheshire, they represent a sample generalisable to the UK primary care population.[29]

This is the first research to examine monthly trajectories of these symptoms in CVD patients.

There are several limitations to our work. First, due to the prospective nature of our research, there was attrition during the follow-up period. Those in the monthly analysis were slightly younger and more likely to be male than all those responding at baseline. Related to attrition, our use of multiple imputation may be limited as we are unable to determine if missing values are truly missing at random. In the monthly questionnaire, our definitions of chest pain and SoB were based on a single question for each. Though using a short monthly questionnaire improved the month-on-month response rates, such a simple definition may limit the accuracy of patient responses. The questions related to chest pain and SoB that occurred during activity; however, it is possible that the symptoms may be non-cardiovascular in origin. We also did not identify the number or duration of symptom episodes other than in how many weeks they had occurred in the last month. However, no matter the specific cause of the chest pain or SoB, patients with CVD are experiencing these symptoms, and therefore these need to be addressed.

## Conclusions

Several modifiable patient characteristics (weight, psychological health and number of other painful body sites) are associated with the experience of both occasional and weekly chest pain and SoB during activity. Future interventions to improve CVD symptom experience may wish to target these covariates. By identifying trajectories of symptoms over time, care may be better tailored to improve outcomes in those patients with CVD at greatest risk of worse symptoms and disease progression.

**Acknowledgements** We are grateful for the participation of general practice teams and their patients in supporting the 2C study. Acknowledgements are given to the Keele survey, network, administration and management teams who supported the study.

**Contributors** Guarantor of overall study integrity: KPJ. Study concept and design: KPJ, JAP and UTK. Data collection and interpretation: LAB, KPJ, JAP and UTK. Statistical analysis: LAB and KPJ. Manuscript preparation: LAB, KPJ, JAP and UTK. Final approval of manuscript: LAB, KPJ, JAP and UTK.

**Funding** LAB is funded by a National Institute for Health Research (NIHR) Research Methods Fellowship, and UTK was funded by an NIHR Post-Doctoral Fellowship (grant no. PAS/PDA/03/07/035). The study sponsors had no role in study design; in the collection, analysis and interpretation of data; in the writing of the report; and in the decision to submit the paper for publication. The views expressed are those of the authors and not necessarily those of the NHS, the NIHR or the Department of Health.

**Competing interests** None declared.

**Ethics approval** Ethics approval for the 2C study was granted by Cheshire Research Ethics Committee (reference number 09/H1017/40).

**Provenance and peer review** Not commissioned; externally peer reviewed.

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
