## [Reviewer comments · BMJ Open]

ARTICLE DETAILS

TITLE (PROVISIONAL)	Chest pain and shortness of breath in cardiovascular disease: a prospective cohort study in UK Primary Care
AUTHORS	Barnett, Lauren; Prior, James A.; Kadam, Umesh; Jordan, Kelvin

VERSION 1 - REVIEW

REVIEWER	John Robson Centre for Primary care and Public Health Queen Mary University of London
REVIEW RETURNED	16-Jan-2017

GENERAL COMMENTS	In general I thought this was a well written paper with aims that were useful and outcomes that appropriately described the aims. My only concern is that the claims in the Discussion that do not follow from the results. line 274 "our findings highlight an opportunity to improve physical symptom experience through mental health. In particular, patients identified with both symptoms may prove the most appropriate group for initial targeting of mental health interventions, particularly in those patients who are optimally managed with anti-anginal medications and heart failure therapy." "We have shown that chest pain and SoB are associated with anxiety and depression and pain elsewhere, and addressing these characteristics may benefit both symptoms" 1. There are few references on the well recognised association of anxiety and depression to chest pain and SOB in people with CVD though there are numerous studies - eg below Clin Res Cardiol. 2013 Aug;102(8):571-81. doi: 10.1007/s00392-013-0568-z. Epub 2013 May 1. Persistent angina: highly prevalent and associated with long-term anxiety, depression, low physical functioning, and quality of life in stable angina pectoris. Jespersen L1, Abildstrøm SZ, Hvelplund A, Prescott E. Int J Cardiol. 2016 Dec 23. pii: S0167-5273(16)34581-8. doi: 10.1016/j.ijcard.2016.12.091. [Epub ahead of print] Depression and chest pain in patients with coronary artery disease. Hayek SS1, Ko YA2, Awad M1, Del Mar Soto A1, Ahmed H1, Patel K1, Yuan M3, Maddox S4, Gray B1, Hajjari J4, Sperling L1, Shah A5, Vaccarino V5, Quyyumi AA6. 2. Of more concern, no reference is provided for the assertion that
--

	interventions designed to improve either anxiety or depression have been shown to reduce chest pain or SOB in patients with CVD - assertions which are fairly central to the authors views - is it possible to reference these assertions regarding interventions to reduce anxiety or depression in people with CVD and chest pain - or to reword. Similarly, the paper didnt really deal with people who were and were not optimally managed so its a little bit difficult to make recommendations on that basis. Having said that I thought that it was useful to document and group this association of depression and anxiety with these symptoms.
--	--

REVIEWER	Stephen Thielke University of Washington, Seattle, WA
REVIEW RETURNED	30-Jan-2017

GENERAL COMMENTS	This is interesting research about an important topic. Unfortunately it suffers from some fundamental methodological problems. I discuss them in roughly descending order of importance.  1. The ostensible topic is “trajectories”, but the article does not address trajectories at all. Instead, it defines patterns of cross-sectional association at several time points. A trajectory is a specific type of change over time (e.g. improving, worsening, stable). 2. You treat shortness of breath and chest pain as if they were independent. In fact, they likely have a huge amount of overlap. Both are reported by about half of respondents. They show similar patterns of association. They seemed to co-occur across time. Reference #7 justifies associating the two with each other, as well as with other forms of pain. It is absolutely essential to address this finding in the design of the research, and the discussion. Specifically, you need to consider how the two variables might be manifestations of the same underlying phenomenon, with subtle variations. You cannot assume that they are separate predictors. To this end, you should present analyses addressing the cross-sectional and temporal associations of SoB and chest pain. This is missing. For example, you state, “Given the strong association between anxiety and depression, we combined the two scales and defined those with borderline or probable anxiety or depression (based on standard cut-offs) as anxious or depressed.” By the same logic, if SoB and chest pain were strongly associated, shouldn't they be collapsed? I am guessing (based on Reference #7) that these are probably roughly as associated as are anxiety and depression.  3. The cluster analysis was, frankly, confusing. I couldn't make heads or tails of exactly what the clusters. Most seemed to have little face validity. Even sophisticated readers would be unable to make sense of the eight different clusters, and there did not seem to be any general finding about how they were distributed. A more productive analysis besides that of trajectories would be to quantify the association between chest pain and SoB over multiple
---

	time points. What is their concordance or discordance in different patients? As one changes, does the other? While this may be part of the analysis, I did not see it. 4. The discussion about “modifiable risk factors” was unproductive and unjustified. Your work provided no information about which of the factors did, or could, vary across time. Given other research, I imagine that none of these factors is in fact subject to much modification through structured interventions (although they do vary across time). There may be treatments, but how often do they work? 5. The discussion seemed to assume that by changing modifiable risk factors, one could improve chest pain or SoB. There is no evidence of this in the literature, or from your findings. The underlying logic misses the most obvious point, that the risk factors in this study may in fact be caused by SoB or chest pain. For instance, poor psychological health may be a response to pain or dyspnea. Please reconsider this general point in framing your discussion. An example of this that illustrates the erroneous (or at least unsupported) logic is the claim that: “our findings highlight an opportunity to improve physical symptom experience through mental health.” This type of observational research cannot — no matter the strength of the findings — support such a conclusion. Please think through this issue. 6. You grouped together the middle three categories of the SoB and chest pain variables (1-3 weeks). What is your rationale for doing so? 7. The imputation strategy seemed weak. Given the sampling frame, it makes more sense to consider only those who had complete data, and to recognize this as a weakness.
--	--

VERSION 1 – AUTHOR RESPONSE

Reviewer 1

1. There are few references on the well-recognised association of anxiety and depression to chest pain and SOB in people with CVD though there are numerous studies - eg below

Clin Res Cardiol. 2013 Aug;102(8):571-81. doi: 10.1007/s00392-013-0568-z. Epub 2013 May 1. Persistent angina: highly prevalent and associated with long-term anxiety, depression, low physical functioning, and quality of life in stable angina pectoris. Jespersen L1, Abildstrøm SZ, Hvelplund A, Prescott E.

Int J Cardiol. 2016 Dec 23. pii: S0167-5273(16)34581-8. doi: 10.1016/j.ijcard.2016.12.091. [Epub ahead of print] Depression and chest pain in patients with coronary artery disease. Hayek SS1, Ko YA2, Awad M1, Del Mar Soto A1, Ahmed H1, Patel K1, Yuan M3, Maddox S4, Gray B1, Hajjari J4, Sperling L1, Shah A5, Vaccarino V5, Quyyumi AA6.

• Response: We have now included these references to support our comments on the associations between CVD symptoms and anxiety and depression.

P.13, line 313 “Approximately two-thirds of patients with CVD in the most severe symptom clusters (frequent or persistent chest pain and SoB) had anxiety or depression. These results are supported by longitudinal research which reported poorer treatment outcomes in angina patients with chest pain and depression, compared to those without depression (25) and that in angina patients with persistent

symptoms, long-term anxiety and depression were more likely (26).”

2. Of more concern, no reference is provided for the assertion that interventions designed to improve either anxiety or depression have been shown to reduce chest pain or SOB in patients with CVD - assertions which are fairly central to the authors views - is it possible to reference these assertions regarding interventions to reduce anxiety or depression in people with CVD and chest pain - or to reword.

Similarly, the paper didn't really deal with people who were and were not optimally managed so it's a little bit difficult to make recommendations on that basis.

- Response: We have toned down our language throughout the paper and reworded as per the reviewer's suggestion. We now highlight that the associations we have found between these CVD symptoms and poor mental health present an opportunity for future research to test whether interventions for anxiety or depression would influence the symptoms of chest pain or shortness of breath.

- We have amended the text in the Discussion and conclusion:

P.13, line 318 “Our research shows that the psychological health of CVD patients, especially those with frequent chest pain and SoB is poor. However, at present our findings only indicate the association between chest pain, SoB & poor mental health. It remains unclear whether interventions aimed at anxiety or depression would improve these CVD symptoms, though this does present a potential avenue for future research. “

P.15, line 368 “Several modifiable patient characteristics (weight, psychological health, and number of other painful body sites) influence are associated with the experience of both occasional and weekly chest pain and shortness of breath during activity. Future interventions to improve CVD symptom experience may wish to target these covariates. “

Reviewer 2

1. The ostensible topic is “trajectories”, but the article does not address trajectories at all. Instead, it defines patterns of cross-sectional association at several time points. A trajectory is a specific type of change over time (e.g. improving, worsening, stable).

- Response: Thank you for your comment that this is “...interesting research about an important topic”. With respect, we disagree that this article does not address trajectories. Phase 2 addresses the course of chest pain and shortness of breath over time, taking into account repeated measures of these symptoms. The analysis approach used (“dual trajectory latent class analysis”) explicitly models trajectories. We observed some common trajectories which were stable (patients in cluster 1 reporting no symptoms over the whole 10 months; patients in cluster 7 who had shortness of breath without chest pain in every month and patients in cluster 8 who had chest pain and shortness of breath in every month). Patients in other clusters tended to fluctuate more in symptoms over time, reporting symptoms in few, some, or most months depending on cluster. What we did not observe, probably due to the prevalent nature of this cohort who may have had CVD for some time, were groups of people worsening or improving. We have added to the introduction and discussion.

Introduction: P.4, line 90 “Symptoms of chest pain and SoB are not typically isolated events.

Assessing how patients' symptoms co-occur and change over time may help identify common patterns (trajectories) of these symptoms which may be related to long-term outcomes. However, how the experience of chest pain and SoB in patients with CVD varies over time remains unclear.”

Discussion: P.13, line 297 “Three of our identified clusters consisted of patients with stable conditions. The most common cluster (containing a quarter of our sample) reported no symptoms every month. By contrast, one cluster reported persistent chest pain and SoB every month, and another reported

persistent SoB without chest pain every month. Symptoms of patients in the remaining clusters fluctuated more. We did not observe a cluster who improved or worsened over the 10 months, which may reflect the prevalent rather than incident nature of the cohort.”

2. You treat shortness of breath and chest pain as if they were independent. In fact, they likely have a huge amount of overlap. Both are reported by about half of respondents. They show similar patterns of association. They seemed to co-occur across time. Reference #7 justifies associating the two with each other, as well as with other forms of pain. It is absolutely essential to address this finding in the design of the research, and the discussion.

Specifically, you need to consider how the two variables might be manifestations of the same underlying phenomenon, with subtle variations. You cannot assume that they are separate predictors. To this end, you should present analyses addressing the cross-sectional and temporal associations of SoB and chest pain. This is missing.

For example, you state, “Given the strong association between anxiety and depression, we combined the two scales and defined those with borderline or probable anxiety or depression (based on standard cut-offs) as anxious or depressed.” By the same logic, if SoB and chest pain were strongly associated, shouldn't they be collapsed? I am guessing (based on Reference #7) that these are probably roughly as associated as are anxiety and depression.

- Response: We agree that shortness of breath and chest pains are likely to be highly correlated. This is the exact rationale behind us using a dual trajectory approach as the assumption behind this methodology is that the two variables are correlated and hence they are not treated as independent variables within the modelling procedure. What is interesting from the clusters is that whilst chest pain and shortness of breath do often co-occur, shortness of breath is more common and does occur without chest pain (in over a third of our cohort). We have added to the rationale of using dual trajectories modelling in the methods, and added to the results and to the discussion to emphasize these points.

Methods P.8, line 186 “Dual trajectory Latent Class Growth Analysis (LCGA) was carried out to group (cluster) respondents into the most common trajectories of self-reported chest pain and SoB over the 10 months using the repeated monthly measures of the two symptoms (20). Each cluster, therefore, represents a common trajectory of both chest pain and SoB over the 10 months. A fundamental assumption behind dual trajectory LCGA is that the two symptoms, in this case, chest pain and SoB, are correlated. “

Results P.11 line 267 “Across clusters, in those reporting symptoms, SoB was more common than chest pain. One cluster of patients reported infrequent breathlessness symptoms (reporting SoB for up to 3 weeks on a median of 3 of the 10 monthly questionnaires: n=93, 14%), another cluster of patients reported occasional breathlessness symptoms (reporting SoB for up to 3 weeks in a median of 8 out of the 10 months: n=83, 13%) and a third cluster of patients reported persistent breathlessness symptoms (reporting SoB in all of the last 4 weeks in all monthly questionnaires: n=52, 8%), but people in these three clusters rarely reported chest pain.”

Discussion (Page 14, line 336) “Our clusters provide distinct patterns of symptom experience over time within a CVD population. Three quarters of CVD patients experienced either shortness of breath or chest pain; however, approximately a third of our sample (35%) reported SoB only, either infrequently (14%), occasionally (13%) or persistently (8%). These may represent a cluster of patients at greater risk of poor outcomes. In a large sample of US participants with coronary artery disease at baseline, the mortality rates were twice as high in patients with SoB compared to those with other symptoms and four times greater than in those with no symptoms at presentation (28). Symptoms such as chest pain and SoB may act as prognostic indicators for outcomes resulting in (8), or from CVD (28) and therefore our findings present a range of likely trajectories to which future research may tailor interventions.”

3. The cluster analysis was, frankly, confusing. I couldn't make heads or tails of exactly what the

clusters. Most seemed to have little face validity. Even sophisticated readers would be unable to make sense of the eight different clusters, and there did not seem to be any general finding about how they were distributed.

- Response: We apologise for any lack of clarity around the cluster analysis. We have revised the methods section (page 8, line 185), and results and added further to the discussion of the trajectories (as detailed in responses to the other comments). The attribution of the clusters is clearly stated on p.11, line 257 and in table 5.

A more productive analysis besides that of trajectories would be to quantify the association between chest pain and SoB over multiple time points. What is their concordance or discordance in different patients? As one changes, does the other? While this may be part of the analysis, I did not see it.

- Response: As stated above, we have shown that whilst SoB and chest pain do co-occur, SoB is a more common symptom and frequently occurs without chest pain. A third of respondents reported SoB without chest pain. We have now made this clearer in the results and discussion as detailed in responses to the other comments from this reviewer.

4. The discussion about “modifiable risk factors” was unproductive and unjustified. Your work provided no information about which of the factors did, or could, vary across time. Given other research, I imagine that none of these factors is in fact subject to much modification through structured interventions (although they do vary across time). There may be treatments, but how often do they work?

- Response: Please see our response to Reviewer 1, comment 2. We have reworded the discussion and now highlight that the associations we have found between these CVD symptoms and poor mental health present an opportunity for future research to test whether interventions for anxiety or depression would influence the symptoms of chest pain or shortness of breath

5. The discussion seemed to assume that by changing modifiable risk factors, one could improve chest pain or SoB. There is no evidence of this in the literature, or from your findings. The underlying logic misses the most obvious point is that the risk factors in this study may in fact be caused by SoB or chest pain. For instance, poor psychological health may be a response to pain or dyspnoea. Please reconsider this general point in framing your discussion.

An example of this that illustrates the erroneous (or at least unsupported) logic is the claim that: “our findings highlight an opportunity to improve physical symptom experience through mental health.” This type of observational research cannot — no matter the strength of the findings — support such a conclusion. Please think through this issue.

- Response: As per our response to reviewers 1’s comment, we have toned down our language to state that though associations arise, further research is required to test whether modifying these risk factors will influence CVD symptoms

6. You grouped together the middle three categories of the SoB and chest pain variables (1-3 weeks). What is your rationale for doing so?

- Response: We wished to distinguish more episodic symptoms (1-3 weeks in a month) from no symptoms in a month (“not at all”) and continuous symptoms (“for 4 weeks”). For ease of interpretation, we combined the middle categories concerning episodic symptoms.

We have added to the Methods:

P.6, line 137 “For analysis, these were categorised into: “not at all”, “for 1-3 weeks”, and “for 4 weeks”, representing no, episodic, and persistent monthly symptoms. “

7. The imputation strategy seemed weak. Given the sampling frame, it makes more sense to consider only those who had complete data, and to recognize this as a weakness.

- Response: The nature of the analyses (multilevel modelling and LCGA) meant we could include

responders who did not have complete data for all 10 months. We used responders who had completed at least half the monthly questionnaires for the main analysis. Multiple imputation has recognised limitations, but we have used it here only as a sensitivity analysis to test the robustness of study findings to attrition. As stated in the Discussion (Page 15, line 352), we have acknowledged issues with attrition and the limitation of using multiple imputation where data may not be missing at random.